# Sustainable Treatment for Sulfate and Lead Removal from Battery Wastewater

**Hong Ha Thi Vu [1,2], Shuai Gu [1] , Thenepalli Thriveni [1] , Mohd Danish Khan [3], Lai Quang Tuan [3,4] and Ji Whan Ahn [1,*]**

[1] Center for Carbon Mineralization, Mineral Resources Division, Korea Institute of Geoscience and Mineral Resources, 124 Gwahak-ro, Gajeong-dong, Yuseong-gu, Daejeon 34132, Korea

[2] Faculty of Biotechnology, Chemistry and Environmental Engineering, Phenikaa University, Hanoi 100000, Vietnam

[3] Resources Recycling Department, University of Science and Technology, 217, Gajeong-ro, Yuseong-gu, Daejeon 34113, Korea

[4] Tectonic and Geomorphology Department, Vietnam Institute of Geoscience and Mineral Resources (VIGMR), 67 Chienthang Street, Hadong district, Hanoi 151170, Vietnam

\* Correspondence: ahnjw@kigam.re.kr

**Abstract:** In this study, we present a low-cost and simple method to treat spent lead–acid battery wastewater using quicklime and slaked lime. The sulfate and lead were successfully removed using the precipitation method. The structure of quicklime, slaked lime, and resultant residues were measured by X-ray diffraction. The obtained results show that the sulfate removal efficiencies were more than 97% for both quicklime and slaked lime and the lead removal efficiencies were 49% for quicklime and 53% for slaked lime in a non-carbonation process. After the carbonation step, the sulfate removal efficiencies were slightly decreased but the lead removal efficiencies were 68.4% for quicklime and 69.3% for slaked lime which were significantly increased compared with the non-carbonation process. This result suggested that quicklime, slaked lime, and carbon dioxide can be a potential candidate for the removal of sulfate and lead from industrial wastewater treatment.

**Keywords:** sustainable; battery wastewater; sulfate removal; lead removal

## 1. Introduction

Currently, lead–acid battery is an important industry in the world and has been commonly employed as secondary sources of energy due to its low cost, high energy density, high specific energy, high-rate discharge capability, and safety [1,2]. The lead–acid battery is generally used in vehicles as an energy storage device, backup power supply, and stationary applications [3–7]. However, lead–acid batteries have a fundamental disadvantage of low life expectancy. The life expectancy of the lead batteries is no more than five years. Thus, a huge amount of batteries become expired and are discarded every year. In a spent lead–acid battery recycling plant, the acid electrolyte is regularly gathered and assign to further purification [8]. However, the spent electrolyte is discharged and collected into a tank in a normal disassembling lead–acid battery recycling company thus the used acidic electrolyte cannot recycle by purification because the spent electrolyte was contaminated with pollutants in the tank.

The high lead concentration and low pH, due to the presence of excessive sulfuric acid in the solution perform, comprise the main contaminations of battery wastewater (BWW). The average concentration of lead in wastewater is about 3–15 mg/L and the pH of wastewater falls in the range of 1.6-2.9 [9]. If the battery wastewater is not treated well before discharge to environment, lead can contaminate food and water, and be present in nature. This can cause extreme harm to human health

such as damage to organs, kidney, and nervous systems [10,11]. Moreover, high sulfate concentration in drinking water can be responsible for serious cases of diarrhea and dehydration to human and animals [12]. The industrial wastewater is limited to under 500 mg/L sulfate in the legislation of many countries [13]. Several research have been reported for efficient removal of lead and sulfate from wastewater such as ion exchange [14–16], electrochemical reduction [17], activated carbon [18,19], biosorption [20], adsorption [21–23], and membrane separation [24]. The efficiency of these methods depends on the lead and sulfate concentration, pH, the economics involved, and social factor set in the standard of government agencies. Nevertheless, these methods are expensive and problems related to the disposal of concentrate or regenerating solution. Besides, the chemical precipitation method has low cost, simplicity, and high removal efficiency of pollutants. Of the precipitation methods, the hydroxide precipitation is the most widely used method. This process involves increased pH to convert heavy metal ions to their respective hydroxides which can be easily removed by filtration [25–28].

This research proposes a low-cost method for treating spent lead–acid battery wastewater by quicklime and slaked lime which are generally cheap due to their abundance in nature. The precipitation method is used to efficiently remove sulfate and lead from the wastewater. In addition, carbon dioxide gas was bubbled into the reaction to increase lead removal efficiencies as well as reduce the pH value to about 7 to meet relevant standards of environmental regulations. Furthermore, the utilization of $CO_2$ in the carbonation method also contributes to the international targets on greenhouse gas reduction.

## 2. Materials and Method

### 2.1. Materials

The quicklime CaO and slaked lime $Ca(OH)_2$ were collected from a company in Republic of Korea. The composition of quicklime and slaked lime are shown in Table 1. In both samples, the dominant component is CaO with 91.25 wt % for quicklime and 66.63 wt % for slaked lime. The raw lead–acid battery wastewater sample was generated from a lead–acid battery company and kept in plastic bottles. The battery company had no recycling system; therefore, the sulfuric acid from the used lead–acid battery was directly poured into a storage tank. The main contaminated compositions in the wastewater were sulfate and lead (Table 2). Carbon dioxide ($CO_2$) used was of 99.9% purity and obtained from Chungang Gas Company in Republic of Korea.

**Table 1.** Chemical composition (by mass) of quicklime and slaked lime (wt %).

| Compound Name | $SiO_2$ | $Al_2O_3$ | $Fe_2O_3$ | CaO | MgO | $K_2O$ | $Na_2O$ | $TiO_2$ | MnO | $P_2O_5$ | Ig.Loss |
|---|---|---|---|---|---|---|---|---|---|---|---|
| Quicklime (CaO) | 1.98 | 0.31 | 0.38 | 91.25 | 1.27 | 0.05 | <0.02 | 0.02 | 0.02 | 0.01 | 4.75 |
| Slaked lime $Ca(OH)_2$ | 1.84 | 0.33 | 0.40 | 66.63 | 3.36 | 0.08 | <0.02 | 0.02 | 0.02 | 0.02 | 26.96 |

Ig.Loss: Loss on ignition.

**Table 2.** Spent lead–acid wastewater compositions.

| Parameter | Value |
|---|---|
| Sulfate (mg/L) | 147,000 |
| Lead (mg/L) | 3.01 |
| pH | −0.52 |

### 2.2. Experiment

The experiments were carried out in a 500 mL reactor with two-wings-oar-type impeller rotating at 400 rpm at room temperature. A glass electrode of pH meter was dipped into solution in the reactor for measuring pH of the mixture during a reaction. Then, 200 mL raw spent lead–acid battery wastewater was slowly added into the 500 mL reactor containing 200 mL of calcium oxide CaO with concentration

2.6 M under vigorous stirring. The reaction was kept stirred until a homogenous mixture was obtained at pH 12 and 90 minutes for precipitation of the $SO_4^{2-}$. In order to separate residual sludge from residual solution, the suspension was filtered under vacuum using a paper filter, and then the solid residue was washed several times by distilled water. The residual solution was further filtered through 0.45 μm Hyundai Micro syringe filter and was diluted to 500 mL with distilled water for further ion chromatography and inductively coupled plasma-optical emission spectrometry measurements' purpose. The solid sludge was dried in an oven at 100 °C for overnight. The slaked lime ($Ca(OH)_2$) sample was also mixed with battery wastewater using a similar methodology to compare results with quicklime.

Unlike the non-carbonation process, with carbonation, when reaction reached stable pH (about pH 12), carbon dioxide ($CO_2$) was injected into the reaction system via porous bubbler with 1 L/min flow rate to further stabilize lead. When the pH was reduced to a range of 6–7 the reaction was stopped. The solid and solution residues were collected to analysis as same way with the non-carbonation process.

The residual sulfate and lead were analyzed by ion chromatography and inductively coupled plasma-optical emission spectrometry. The contamination removal efficiency *E* (%) was determined using the following equation:

$$E\% = \frac{C_0 - C}{C_0} \times 100$$

where $C_0$ is the primary concentration of total sulfate or lead (mg $L^{-1}$) and *C* is the concentration of total sulfate or lead after treatment (mg $L^{-1}$).

*2.3. Physical Characterization*

The chemical composition of quicklime and slaked lime samples were performed using X-ray fluorescence spectrometer (XRF, Shimadzu, Japan). X-ray diffraction (XRD, D/MAX 2200, Rigaku, Japan) with X-ray source of Cu Kα (λ = 0.15406 nm) in the scan range of diffraction angle 2θ from 10° to 90° was used to identify the crystallization of the lime (CaO), quick lime, and solid residues samples. The pH was measured by Orion Versa Star Pro (Thermo Scientific, USA) pH meter with a glass electrode. Before measuring, the pH meter was calibrated by different buffer solutions for an accurate pH. The concentration of sulfate before and after treatment was measured by ion chromatography (ICS-3000, Dionex). Lead concentration analysis was determined using an inductively coupled plasma-optical emission spectrometry (ICP-OES, Optima 8300, PerkinElmer).

## 3. Results and Discussion

The pH value of the suspensions indicates the progress of the reaction. In the non-carbonation process, the reaction reaches chemical equilibrium when the pH is stable in the solution. Initially, the pH was increased rapidly within 5 minutes and then increased slowly until equilibrium was attained. The pH surpasses 12 after 90 minutes reaction therefore the sulfate ions were removed and $OH^-$ ion from excessive calcium hydroxide ($Ca(OH)_2$) in the mixture (Figure 1a,c).

In the carbonation process, the mixtures reached equilibrium after 90 minutes of reaction time. Then $CO_2$ gas was injected into the suspension through porous bubbler. The pH value of the suspension decreased because hydroxide $OH^-$ ion was reduced due to the reaction between carbon dioxide with calcium hydroxide and calcium carbonate was formed (Figure 1b,d). When the pH of the mixture was about 7, which was suitable for a number of applications, carbon dioxide gas was stopped being injected into the solution.

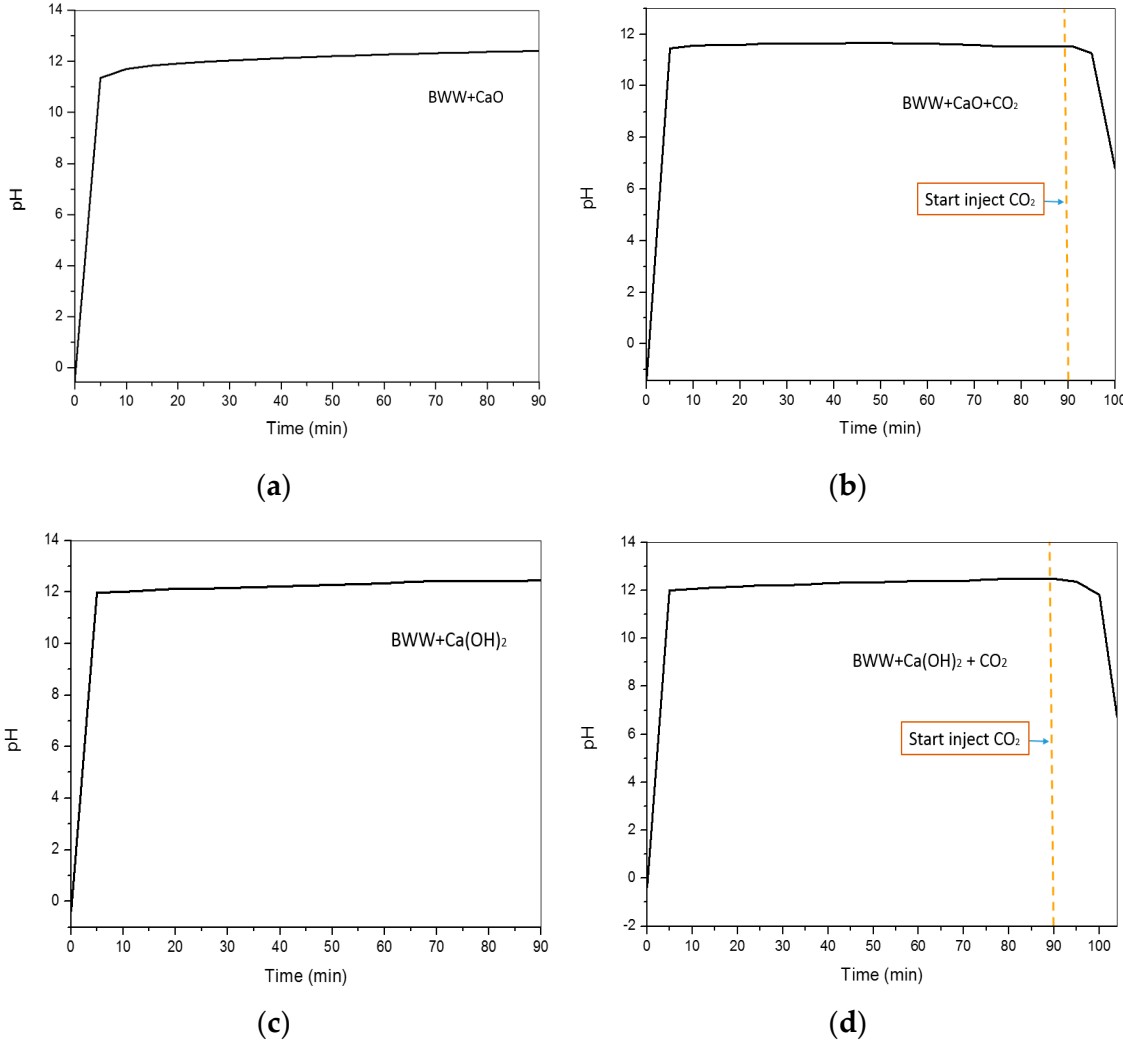

**Figure 1.** Changes in pH values at different time for (**a**) battery wastewater (BWW) + quicklime, (**b**) battery wastewater + quicklime + $CO_2$, (**c**) battery wastewater + slaked lime, and (**d**) battery wastewater + slaked lime + $CO_2$.

Figure 2 shows XRD patterns of quicklime and slaked lime raw samples. The raw quicklime is highly crystalline and all the observed diffraction peaks are good agreement with cubic lime phase CaO of space group Fm-3m (PDF#99-0070) (Figure 2a). The XRD peaks at $2\theta$ = 32.2°, 37.3°, 53.8°, 64.1°, 67.3°, 79.6°, and 88.5° corresponded to (111), (200), (220), (311), (222), (400), and (331) plane of calcium oxide phase, respectively. The dominant peak is observed at (200) plane. For the slaked lime case, the major XRD peaks resembled very well the characteristic peaks of portlandite phase with space group P-3m1 (PDF#87-0673) [29]. The peaks at 18.0°, 28.7°, 34.1°, 47.1°, 50.8°, 54.3°, 56.1°, 62.6°, 64.2°, and 77.7° $2\theta$ were assigned to the (001), (100), (101), (102), (110), (111), (003), (021), (013), and (004) plane of hexagonal portlandite $Ca(OH)_2$ phase, respectively. Besides, a minor peak was detected at $2\theta$ = 29.3 which corresponds to (104) plane of rhombohedral calcite phase with space group R-3c since the portlandite $Ca(OH)_2$ reacted with $CO_2$ gas from surrounding air and calcite formed [30].

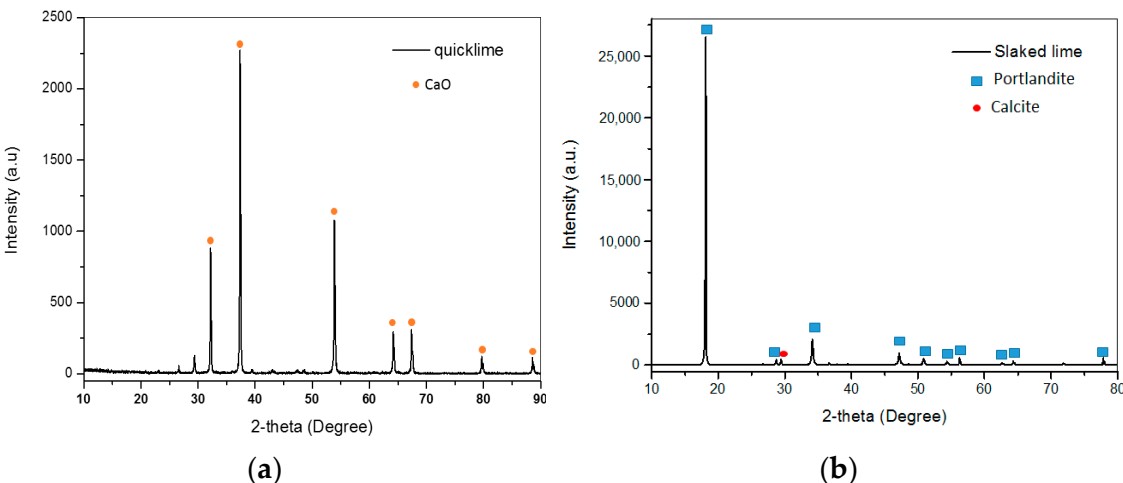

**Figure 2.** X-ray diffraction (XRD) patterns of (**a**) quicklime and (**b**) slaked lime.

Regarding the identification of crystal structure of solid residues after reaction, the samples was investigated by X-ray diffraction, as shown in Figure 3. Figure 3a,c show the XRD patterns of white precipitation of the non-carbonation process. Four substances can be observed which are gypsum, calcium hemihydrate ($CaSO_4(H_2O)_{0.5}$), calcium hydroxide ($Ca(OH)_2$), and calcium sulfate ($CaSO_4$), respectively. The precipitates were presented portlandite $Ca(OH)_2$ owing to the excessive calcium hydroxide from quicklime or slaked lime in the mixtures.

For the carbonation process of quicklime and slaked lime, gypsum, calcite and calcium sulfate substances were formed during the precipitation process and they can be found in the XRD peaks (Figure 3b,d). The calcium hydroxide was not found in this case due to complete carbonation.

In addition, the XRD patterns of solid residues of both the non-carbonation process and the carbonation process could not detect any peaks related to lead precipitates, such as $Pb(OH)_2$ and $PbCO_3$, because the lead precipitate concentrations may be a very small amount in the solid residues.

The removal of sulfate was dominated by calcium sulfate ($CaSO_4$) precipitation. The results showed that more than 97% sulfate removal efficiencies within 90 minutes contact time were achieved in both quicklime and slaked lime in non-carbonation process. The sulfate removal efficiencies were 97.87 and 97.95% for quicklime and slaked lime, respectively.

Figure 4 and Table 3 show the removal efficiencies of lead in normal process and carbonation step. The lead removal efficiency of quicklime in the carbonation process was 68.4%, which was 38.7% higher than that of the case without carbonation (49.3%). The removal efficiency of lead in slaked lime was 53.5% for the normal process and it was increased to 69.3% for the carbonation process. Both quicklime and slaked lime had almost the same effects on increasing lead removal efficiency in the carbonation process. Chen et al. report that lead removal efficiency was significantly decreased since $Pb(OH)_2$ was dissolved when the pH of the solution was above 10 [25,28]. For non-carbonation, the pH of the mixture was higher than 12 after reaction due to high concentration of $OH^-$ ion from $Ca(OH)_2$. Therefore, lead hydroxide was dissolved back into the mixture. For the carbonation process, the pH of the solution was about 7 after the reaction, thus $Pb(OH)_2$ was stable. In addition, when $CO_2$ gas was injected into the mixture, at this point the excessive calcium hydroxide was transformed to $CaCO_3$, then lead can be adsorbed on the surface of calcium carbonate and the cerussite $PbCO_3$ precipitate can be generated as lead ion and carbonate ion both exist in the mixture by the following Reactions (1), (2), and (3). For these reasons, the removal efficiency of lead in carbonation process was higher than that in the non-carbonation process.

$$CO_2 + H_2O = H_2CO_3 \tag{1}$$

$$H_2CO_3 = 2H^+ + CO_3{}^{2-} \tag{2}$$

$$Pb^{2+} + CO_3{}^{2-} = PbCO_3(s) \tag{3}$$

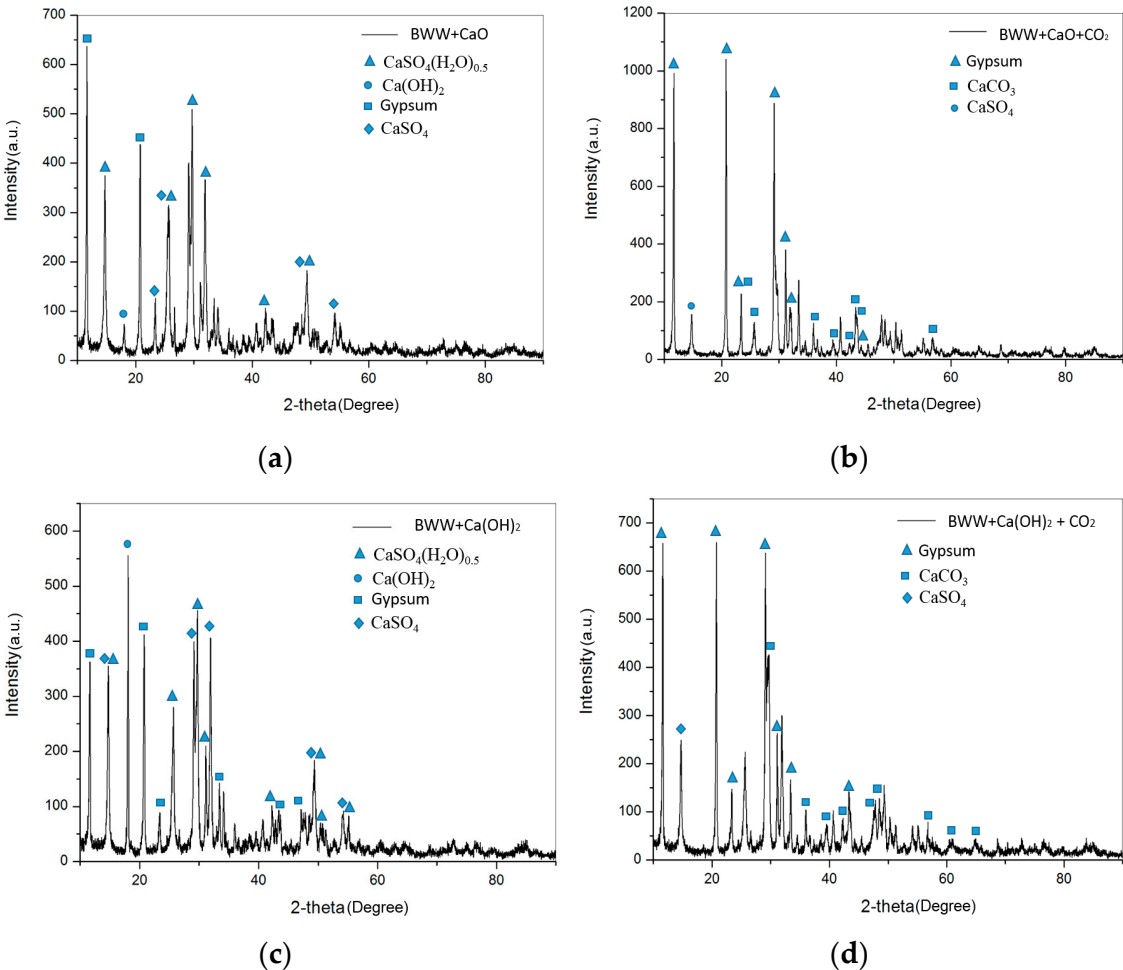

**Figure 3.** XRD patterns of (**a**) solid residue of battery wastewater + quicklime, (**b**) solid residue of battery wastewater + quicklime + $CO_2$, (**c**) solid residue of battery wastewater + slaked lime and (**d**) solid residue of battery wastewater + slaked lime + $CO_2$.

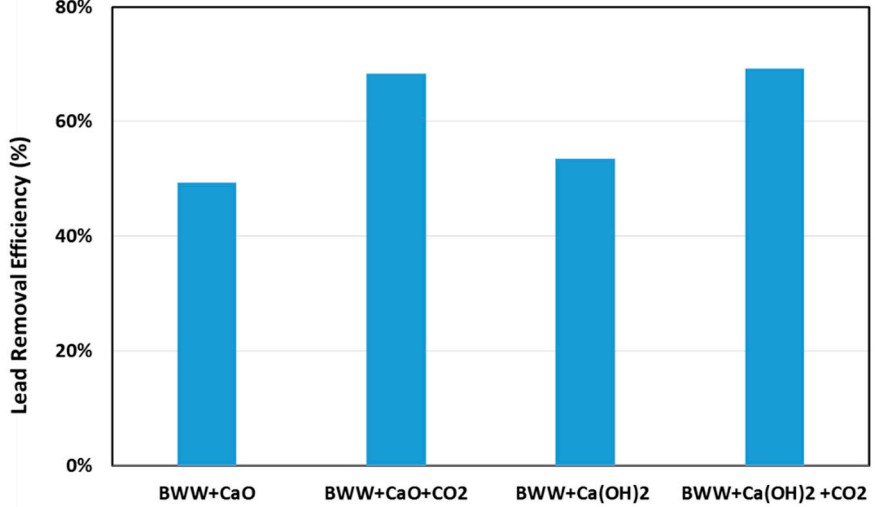

**Figure 4.** Lead removal efficiency with and without carbonation process.

**Table 3.** Sulfate and lead removal efficiencies with different experiments.

| Experiment | Lead Removal Efficiency |
| --- | --- |
| BWW + CaO | 49.3% |
| BWW + CaO + $CO_2$ | 68.4% |
| BWW + Ca(OH)$_2$ | 53.5% |
| BWW + Ca(OH)$_2$ + $CO_2$ | 69.3% |

BWW: Battery wastewater.

## 4. Conclusions

In this work, the sulfate and lead were successfully treated by using quicklime and slaked lime through a precipitation method. For both quicklime and slaked lime samples, the sulfate $SO_4^{2-}$ removal efficiencies were more than 97% and the lead removal efficiencies were 49% for quicklime and 53% for slaked lime. The removal efficiency of lead was increased after using a carbonation step with 68% for quicklime and 69% for slaked lime. The carbonation process not only enhanced the lead removal efficiency in the battery wastewater but also reduced pH to meet requirements of environmental regulations. In addition, the carbonation method is an environment-friendly method since the technology can utilize $CO_2$ generated from industry. Due to the environmental benefit, this carbonation technology is recommended as a simple and low-cost method for industrial wastewater treatment.

**Author Contributions:** H.H.T.V. and S.G. planned and designed the experiment; H.H.T.V. and L.Q.T. carried out the experiments; H.H.T.V. and M.D.K. analyzed the data and wrote the paper; T.T and J.W.A. reviewed and revised the manuscript.

**Acknowledgments:** This research was supported by the National Strategic Project-Carbon Mineralization Flagship Center of the National Research Foundation of Korea (NRF) funded by the Ministry of Science and ICT (MSIT), the Ministry of Environment (ME), and the Ministry of Trade, Industry and Energy (MOTIE).(2017M3D8A2084752).

**Conflicts of Interest:** The authors declare no conflicts of interest.

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
