# Peer review of "Sustainable Treatment for Sulfate and Lead Removal from Battery Wastewater"

_sustainability, doi:10.3390/su11133497_

Reviewer 1 Report

Dear Authors

We believe that your paper is very interesting and your results contribute to a better understanding of the potential effectiveness of some treatments used the removal of pollutants in some industrial wastewater. Overall the paper is clear, however we consider that the introduction could include some information on the precipitation method in the removal of pollutants, in particular on precipitation by hydroxides. Also in the section of the material and methods, Table 2 indicates a negative pH value and I did not understand this value. Section 3 is entitled Results, however, it includes the discussion and, therefore, I suggest that it be called Results and discussion.

Best Regards,

Conceição Mesquita

Author Response

Response to Reviewer 1 Comments

General comments

We believe that your paper is very interesting and your results contribute to a better understanding of the potential effectiveness of some treatments used the removal of pollutants in some industrial wastewater.

Ans) Thank you very much for your comments and encouragement.

Specific comments

Overall the paper is clear, however, we consider that the introduction could include some information on the precipitation method in the removal of pollutants, in particular on precipitation by hydroxides. Also in the section of the material and methods, Table 2 indicates a negative pH value and I did not understand this value. Section 3 is entitled Results, however, it includes the discussion and, therefore, I suggest that it be called Results and discussion.

Ans) Thank you for your useful comment. The introduction has already included some information on the hydroxides precipitation method in the removal of pollutants (line no. 56-60).

 The pH is a negative value due to the very high concentration of acid sulphuric (H2SO4) in the battery wastewater.  According to analysis the concentration of SO42- is 147000 mg/L. In theory, assume all S in the solution is in the form of H2SO4, the concentration of H2SO4 can be calculated C[H2SO4]= 1.53mol/L, so [H+] = 2 x 1.53 = 3.06 and pH = -log[H+] = -0.486. In the battery wastewater, the pH value is very close to the theoretical value.

And the “Results” changed to “Results and discussion” (line no. 117).

Reviewer 2 Report

SUSTAINABILITY

TITLE: Sustainable Treatment for Sulfate and Lead Removal from Battery Wastewater

Dear Authors,

At the beginning I would like to praise you for choosing the real system for your investigation, but unfortunately in my opinion investigation is not conducted on right way. Missing experiments and scientific explanations.

So, my decision is manuscript rejection in this form, after you upgrade the manuscript you can resubmit it.

My explanations follow:

In my opinion Authors more explore removal of sulphate anions and XRD analysis of precipitates which is of lesser importance in this paper than more interesting topics, lead removal.

Why Authors used pH 12 for lead removal and contact time of 90 min?

At pH 10 dominant species is Pb(OH)2 around 75%, while at pH 7, dominate species are Pb2+ (65%) and PbOH+(35%). Also solubility of Pb(OH)2 is the smallest at pH 10.

Concerning PbCO3, dominant species at pH 8 and at same pH is the smallest solubility.

Regarding this, I think that Authors should be examine the effect of pH (pH=9,10,11,12) on lead removal efficiency since to achieve a higher pH requires more lime (higher costs).

Also, did Authors determined the lead concentration temporary in some time intervals within 90 min? Is 90 min necessary or less?

Is really the introduction of CO2 gas contributing to increasing in efficiency? What about dissolving of Pb(OH)2 and also at pH 6-7 dissolving of PbCO3 occurred.

In my opinion is better to conduct experiment with some lime (quicklime or slaked lime) at optimal pH and contact time, separated solid from liquid and then liquid phase neutralised with CO2. On this way sulphate ions are neutralised. From Fig 4 CO2 addition have not positive influence on sulphate reduces.

You have not compared the lead and sulphate concentrations with legal regulations - after treatment, did you decrease the concentration of these ions below the permissible limits for discharging into receiving systems.

What about sludge after treatment? Possible sludge management?

For real application, it is cheaper to use technical lime than the analytical purity, so this is also one of possibility to compare results and improvement manuscript.

TITLE: Sulfate change to Sulphate

ABSTRACT

Line 16-18: “In order to reduce lead concentration from spent lead-acid battery wastewater to meet legislative requirements for discharge, carbon dioxide was introduced into the suspension.” – The sentence in this form does not belong in Abstract section.

INTRODUCTION

The introduction makes sense but is unrelated, one part does not follow the other one. Systematize! E.g. in line 41, The crossing from one to the other sentence is sharp, unrelated.

Line 35-37: “However, the used electrolyte is discharged and collected into a tank in a normal disassembling lead-acid battery recycling company thus the used acidic electrolyte cannot recycle by purification.” – This sentence is not clear for me.

MATERIALS AND METHOD

Line 61: CaO and Ca(OH)2, both or no one in brackets

Make up your mind, mL or ml in all manuscript

RESULTS

Commented before.

CONCLUSIONS

There are no significant conclusions.

Author Response

Please kindly check the attached file.

Thank you very much

Reviewer 3 Report

The paper describes an easy and simple method to remove sulfate and lead from battery wastewater. Paper is mainly well written.

Abstract: It would be good to inform the removal-% of lead in carbon dioxide treatment in lines 21-22 since one aim of abstract is to find new readers (and new citations).

Introduction is good.

Materials and methods:

How many parallel experiments you did? What about determinations?

Line 74 omit And.

Line 81 open ICP as you do it in part 2.3.

Results: Your figures are clear. In line 116 open BWW, since the figures and tables should be independent so that the reader should understand the idea without reading the text. You have it in line 39.

Think if the Figures 4 and 5 would be more informative as tables or as one table?      

Line 170 replace Which as This

Line 176 omit And  (it is not good to start with And).     

Conclusions are clear.  Anyhow, would this method serve also mining industry?

References

In reference 6 there must be a number of paper. The number starts with 01 – 12.

The references 14 and 24 refer to the same paper. In 14 you have all authors but the reference method is not correct. In 24 you have only the three first authors and the rest are lacking. Correct then all numbers from 24.   

Author Response

Response to Reviewer 3 Comments

General comments

The paper describes an easy and simple method to remove sulfate and lead from battery wastewater. Paper is mainly well written.

Ans) Thank you very much for your comments and encouragement.

Abstract: It would be good to inform the removal-% of lead in carbon dioxide treatment in lines 21-22 since one aim of the abstract is to find new readers (and new citations).

Ans) Thank you for your useful comments. We added more information about led removal in carbon dioxide treatment in line no. 24-26.

Introduction is good.

Materials and methods:

How many parallel experiments you did? What about determinations?

Ans) I did four parallel experiments [(1) battery wastewater + quicklime, (2) battery wastewater + quicklime + CO2, (3) battery wastewater + slaked lime and (4) battery wastewater + slaked lime + CO2]. 

Line 74 omit And.

Ans) yes omitted in line no. 85

Line 81 open ICP as you do it in part 2.3.

Ans) Yes, opened ICP in line no. 93-94

Results: Your figures are clear. In line 116 open BWW, since the figures and tables should be independent so that the reader should understand the idea without reading the text. You have it in line 39.

Ans) Opened BWW in line no. 128-129 and line no. 145-147

Think if the Figures 4 and 5 would be more informative as tables or as one table? 

Ans) Added Table 3 for more information about Figure 4.     

Line 170 replace Which as This

Ans) This part is removed

Line 176 omit And  (it is not good to start with And).  

Ans) Yes, omitted And in line no. 172

Conclusions are clear.  Anyhow, would this method serve also mining industry?

Ans)  Thank you for your comment. I think this method serves also can use in the mining industry due to the low cost and the environment benefit.

References

In reference 6 there must be a number of paper. The number starts with 01 – 12.

The references 14 and 24 refer to the same paper. In 14 you have all authors but the reference method is not correct. In 24 you have only the three first authors and the rest are lacking. Correct then all numbers from 24.   

 Ans) As per your recommendations, modified the references in the revised manuscript.
